# SELECTIVE UNDERFITTING IN DIFFUSION MODELS

## ABSTRACT

Diffusion models have emerged as the principal paradigm for generative modeling across various domains. During training, they learn the score function, which in turn is used to generate samples at inference. They raise a basic yet unsolved question: *which* score do they actually learn? In principle, a diffusion model that matches the empirical score in the entire data space would simply reproduce the training data, failing to generate novel samples. Recent work addresses this question by arguing that diffusion models *underfit* the empirical score due to training-time inductive biases. In this work, we refine this perspective, introducing the notion of *selective underfitting*: instead of underfitting the score everywhere, better diffusion models more accurately approximate the score in certain regions of input space, while underfitting it in others. We characterize these regions and design empirical interventions to validate our perspective. Our results establish that selective underfitting is essential for understanding diffusion models, yielding new, testable insights into their generalization and generative performance.

## 1 INTRODUCTION

Diffusion models (Sohl-Dickstein et al., 2015; Ho et al., 2020; Song et al., 2020) have recently shown significant success in generating high-quality samples, achieving state-of-the-art performance in tasks like image, video, and audio generation. During *training*, they learn a *score function*–the gradient of the log density of the data distribution convolved with noise–by reconstructing data that has been corrupted with Gaussian noise, a process called *denoising score matching* (Vincent, 2011). During inference, the learned score is applied iteratively to denoise Gaussian noise to clean samples.

Despite these successes, the learned score presents an apparent paradox: if training were perfect, the model would learn the *empirical score function*–the score with respect to the finite training data–and simply replicate the training data at inference, never generating novel samples. Recent work has sought to resolve this paradox by proposing various mechanisms by which diffusion models *underfit* the empirical score, due to inductive bias or smoothness of neural networks (Kamb & Ganguli, 2024; Niedoba et al., 2024b; Scarvelis et al., 2023; Pidstrigach, 2022; Yoon et al., 2023; Yi et al., 2023).

In this work, we argue that understanding diffusion models requires us to investigate not just *how* they underfit the empirical score, but *where*. Our starting point is the mismatch between the distribution of training samples versus the distribution of inputs encountered during inference trajectories. Consider Figure 1: during training, for most timesteps, samples concentrate in thin shells around each data point (blue shells). Within these shells, the learned score simply points back to the shells' centers, i.e., the training images (green arrows). In contrast, we find that during inference, denoising trajectories quickly land in regions *beyond* these shells (red points), *which were effectively never supervised with the empirical score at training time*.

This observation prompts us to distinguish two distinct regions in the data space: the **supervision region**, where the model is supervised to approximate the empirical score during training, and the **extrapolation region**, where the model receives no supervision but is queried at inference. Our scaling experiments show that as models improve, their approximation of the empirical score *inside* the supervision region improves, while it worsens *outside* (underfitting). In other words, underfitting does not occur in the supervision region–a property we term **selective underfitting**, which, as we demonstrate, **is essential for understanding diffusion models.**

We leverage selective underfitting to design a suite of controlled experiments that shed new light on diffusion models' **(1) generalization**, showing that the *size of the supervision region* strongly affects a model's ability to generalize–enlarging this region degrades generalization, an effect overlooked

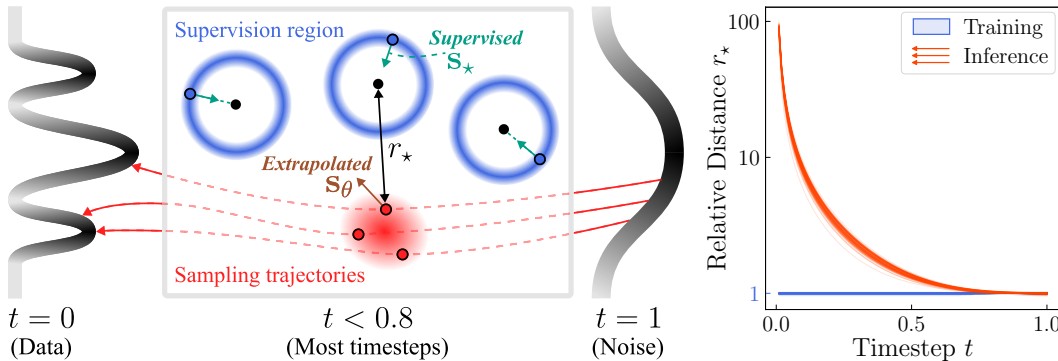

Figure 1: **Selective underfitting.** (a) Training distribution concentrates in a small region of data space, where the model learns the empirical score function $\mathbf{s}_\star$. At inference, sampling trajectories go beyond this region, where predictions are not directly *supervised* and must be *extrapolated*. (b) The plotted distances reflect how far sampling trajectories are from the supervision region.

in prior studies–and **(2) generative performance**, where our novel scaling law between generative performance and *supervision loss* enables directly quantifying the model's ability to *transfer* the supervised score to the extrapolation region. This framework not only clarifies the benefits of recent advances such as REPA (Yu et al., 2024) and diffusion transformers (Peebles & Xie, 2023; Ma et al., 2024), but also suggests a general principle for designing better training recipes. Taken together, our findings establish *selective underfitting* as a central principle for understanding diffusion models.

**Contribution. (1)** We demonstrate *selective underfitting* in diffusion models across two theoretically grounded regions of data space, with empirical validation that refines prior, oversimplified views. Leveraging selective underfitting: **(2)** We advance a mechanism for how diffusion models generate samples beyond their training data that is agnostic to architectural inductive biases, unlike previous theories. **(3)** We propose a novel scaling law to quantify the efficiency of training recipes for diffusion models, clarifying why recent seminal approaches work and providing a unified perspective for designing efficient recipes.

## 2 BACKGROUND: WHAT AND WHERE DO DIFFUSION MODELS LEARN?

In this section, we review the formulation of diffusion models along with the commonly held interpretation, which we refer to as *global underfitting*. This sets the stage for our contrasting viewpoint–*selective underfitting*–introduced in the following sections.

Diffusion models[1] define a stochastic forward process that transforms a data distribution into a Gaussian distribution over timesteps $t \in [0, 1]$: $\mathbf{z}_t = \alpha_t \mathbf{x} + \sigma_t \boldsymbol{\epsilon}$ where $\mathbf{x}$ is a clean data and $\boldsymbol{\epsilon} \sim \mathcal{N}(\mathbf{0}, \boldsymbol{I})$ for schedules $\alpha_t, \sigma_t \colon [0, 1] \to \mathbb{R}$. The models are trained through denoising score matching (DSM, Vincent (2011)) to learn the score function–the gradient of the log density–which is then used at inference to iteratively denoise Gaussian noise into a sample from the data distribution.

**Learning Problem.** Given training data $\mathcal{D} = \left\{ \mathbf{x}^{(i)} \right\}_{i=1}^{N}$, the DSM objective at timestep $t$ is:

$$\mathcal{L}_{\text{DSM}}(t) := \mathbb{E}_{\mathbf{x}^{(i)} \sim \text{Unif}(\mathcal{D}), \boldsymbol{\epsilon} \sim \mathcal{N}(\mathbf{0}, \boldsymbol{I})} \left\| \mathbf{s}_{\boldsymbol{\theta}}(\mathbf{z}_t, t) + \boldsymbol{\epsilon}/\sigma_t \right\|^2 \tag{1}$$

The DSM nature of diffusion model training can be decomposed into two components: (i) *what function* does the model aim to learn? (i.e., what is the minimizer of $\mathcal{L}_{\text{DSM}}$?), and (ii) *where in the data space* does the model learn during training? (i.e., what is the distribution of $\mathbf{z}_t$?). For (i), there exists an analytic optimal solution for Equation (1):

$$\mathbf{s}_\star(\mathbf{z}_t, t) = \frac{1}{\sigma_t^2} \left[ -\mathbf{z}_t + \alpha_t \sum_{i=1}^{N} \text{softmax} \left( -\frac{\|\mathbf{z}_t - \alpha_t \mathbf{x}^{(i)}\|^2}{2\sigma_t^2} \right) \mathbf{x}^{(i)} \right]. \tag{2}$$

Since this is the score function of the empirical training data, we refer to $\mathbf{s}_\star$ as the *empirical score function*, often also called the analytic score function. For (ii), a model is trained on

---

[1]Throughout this work, we use "diffusion" as an umbrella term for both diffusion and flow matching (Lipman et al., 2022; Liu et al., 2022), as they are equivalent (Gao et al., 2025). All discussions apply to both.

noisy versions of the training data, which can be written as a mixture of Gaussians: $\hat{p}_t(\mathbf{z}_t) = \frac{1}{N}\sum_{i=1}^{N}\mathcal{N}(\mathbf{z}_t; \alpha_t\mathbf{x}^{(i)}, \sigma_t^2\mathbf{I})$. Given (i) and (ii), Equation (1) is equivalent to:

$$\mathcal{L}_{\text{DSM}}(t) = \mathbb{E}_{\mathbf{z}_t \sim \hat{p}_t} \|\mathbf{s}_{\boldsymbol{\theta}}(\mathbf{z}_t, t) - \mathbf{s}_{\star}(\mathbf{z}_t, t)\|^2 + (\text{constant}). \tag{3}$$

In principle, $\hat{p}_t$ *has support over the entire data space* for any $t > 0$, since it is Gaussian. This leads to the following common understanding of the learning problem:

> **Perspective A** (Common). *Diffusion models are supervised to (i) approximate the empirical score function $\mathbf{s}_{\star}$, (ii) over the entire data space.*

Perspective A, which we call **global underfitting**, has been widely adopted to study diffusion models, for example, in the context of their generalization (Kamb & Ganguli, 2024; Kadkhodaie et al., 2023; Scarvelis et al., 2023; Yi et al., 2023; Wang et al., 2024a) and memorization (Niedoba et al., 2024b). These works primarily focus on (i), analyzing *how* a neural network globally underfits the empirical score function. In contrast, we argue that (ii)–*where* learning occurs–is the critical component, as underfitting occurs *selectively* on certain regions of the data space.

## 3 RETHINKING WHERE DIFFUSION MODELS UNDERFIT

In this section, we challenge the standard Perspective A from Section 2 that diffusion models learn globally over the entire data space. Instead, we decompose the diffusion model learning into two regions: they are 'supervised' on a restricted region, and forced to 'extrapolate' beyond this region during inference. To motivate this, we begin with a paradox arising from Classifier-Free Guidance (CFG, Ho & Salimans, 2022), which Perspective A cannot explain.

**Example 3.1** (CFG Paradox). CFG jointly trains conditional & unconditional models $\mathbf{s}_{\boldsymbol{\theta}}(\cdot,\cdot,c)$ & $\mathbf{s}_{\boldsymbol{\theta}}(\cdot,\cdot,\varnothing)$, and samples with the combined CFG score to improve sample quality:

$$\mathbf{s}_{\text{CFG}}^{\omega} := \mathbf{s}_{\boldsymbol{\theta}}(\mathbf{z}_t, t, c) + (\omega - 1)[\mathbf{s}_{\boldsymbol{\theta}}(\mathbf{z}_t, t, c) - \mathbf{s}_{\boldsymbol{\theta}}(\mathbf{z}_t, t, \varnothing)]$$

CFG works because the *difference between conditional and unconditional scores* is recognizable; if these scores are identical, CFG reduces to standard conditional sampling. Indeed, these two learned scores *differ meaningfully during inference* (Figure 2, red line). According to Perspective A, both learned scores should globally approximate their empirical counterparts on conditional and unconditional training data: $\mathbf{s}_{\boldsymbol{\theta}}(\mathbf{z}_t, t, c) \approx \mathbf{s}_{\star}^c(\mathbf{z}_t, t)$ and $\mathbf{s}_{\boldsymbol{\theta}}(\mathbf{z}_t, t, \varnothing) \approx \mathbf{s}_{\star}^{\varnothing}(\mathbf{z}_t, t)$. Measuring the difference between the empirical scores, however, $\|\mathbf{s}_{\star}^c - \mathbf{s}_{\star}^{\varnothing}\|$ over $\mathbf{z}_t \sim \hat{p}_t$, we find it is *vanishingly small during training* for most timesteps (Figure 2, blue line). This leads to a paradox: why is there such a discrepancy?

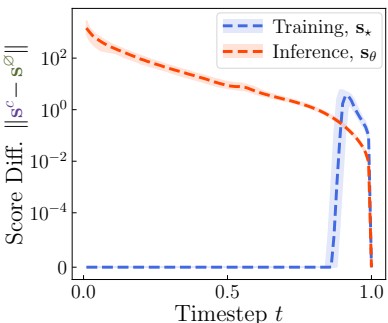

Figure 2: **Difference between conditional and unconditional scores,** measured separately during training and inference. The gap is significant at inference but nearly zero during training.

The paradox prompts us to revisit Perspective A, revealing that the empirical score function during training does not directly translate to the learned score function used at inference. To address this, we now present an overview of our alternative perspective, which naturally resolves this paradox.

**Overview.** Our perspective, illustrated in Figure 1a, is built on the insight that the score function used at inference is not simply a direct neural network approximation of the empirical score function everywhere (Perspective A), but it is determined by (1) the specific region of the data space where the model is supervised (blue shells), which in turn shapes (2) how the model extrapolates outside this region (red lines). We call this **selective underfitting**, as underfitting turns out to occur only in the latter region, as we demonstrate in Section 4.1.

> **Perspective B** (Ours). *Diffusion models are supervised to (i) approximate the empirical score function $\mathbf{s}_{\star}$, (ii) over a restricted region, and extrapolate beyond this region during inference.*

**Resolving the CFG Paradox.** During training, the supervised scores for the conditional and unconditional models are nearly identical. The key difference lies in the regions where supervision is applied: the conditional model is supervised on regions formed by *class-wise* data, while the un-

conditional model is supervised on regions formed by *all* data. Adapting Perspective B, these mismatched supervision regions induce distinct extrapolation (Figure 2), which explains the observed divergence between the two scores during inference.

In the following sections, we formally describe our Perspective B: Section 3.1 provides a theoretical basis for the restricted nature of the supervision region, and Section 3.2 empirically demonstrates extrapolation beyond this region during inference.

### 3.1 TRAINING DISTRIBUTION IS HIGHLY RESTRICTED

In this section, we theoretically justify the following central claim:

**C1.** *Diffusion models are effectively supervised only within a restricted region of data space.*

As reviewed in Section 2, the distribution employed at training is the mixture of Gaussians $\hat{p}_t = \frac{1}{N} \sum_{i=1}^{N} \mathcal{N}(\mathbf{z}_t; \alpha_t \mathbf{x}^{(i)}, \sigma_t^2 \boldsymbol{I})$. In high-dimensional space, these samples concentrate on a set of thin spherical shells, since for a standard Gaussian vector $\boldsymbol{\epsilon} \sim \mathcal{N}(\mathbf{0}, \boldsymbol{I}_d)$, we have $\|\boldsymbol{\epsilon}\| = \sqrt{d}(1 + \mathcal{O}(1))$ with high probability. Consequently, each component of $\hat{p}_t$ concentrates almost all of its mass on a shell of radius $\sigma_t \sqrt{d}$ centered at $\alpha_t \mathbf{x}^{(i)}$, leading to the proposition below (proof in Appendix B.1):

**Proposition 3.2** (Supervision region). *Let $\delta \in (0, 1)$, empirical data $\{\mathbf{x}^{(i)}\}_{i=1}^{N}$, and $\mathbf{z}_t \sim \hat{p}_t$. Then*

$$\mathbb{P}(\mathbf{z}_t \in \mathcal{T}_t(\delta)) \geq 1 - \delta, \quad \mathcal{T}_t(\delta) := \left\{ \mathbf{z} : \exists i \in [N], \left| \|\mathbf{z} - \alpha_t \mathbf{x}^{(i)}\| - \sigma_t \sqrt{d} \right| \leq \sigma_t \sqrt{d \log(1/\delta)} \right\}.$$

We call $\mathcal{T}_t(\delta)$ the *supervision region*: the union of thin spherical shells around the data points that, taken together, contain $\mathbf{z}_t \sim \hat{p}_t$ with high probability. In practice, these shells are non-overlapping for most time steps because each shell's radius $\sigma_t \sqrt{d}$ is much smaller than the separation between their centers, i.e., $\sigma_t \sqrt{d} \ll \min_{i,j} \|\alpha_t \mathbf{x}^{(i)} - \alpha_t \mathbf{x}^{(j)}\|$. We quantify this separation using the *Bhattacharyya coefficient* (Bhattacharyya (1943), Appendix B.2)—a standard measure of distributional overlap—computed on the ImageNet dataset (Deng et al., 2009) and in the latent space of the SD-VAE (Rombach et al., 2022). Figure 4 (right) shows this overlap coefficient across time; it is essentially zero for $t \in [0, 0.8]$, indicating negligible overlap in this regime.

**Empirical Score Collapses.** The concentration and separation results reveal more about the empirical score. With high probability, $\mathbf{z}_t \sim \hat{p}_t$ lies in some $i$-th shell and is far from all others, making the softmax weights in Equation (2) extremely imbalanced: nearly 1 on $\mathbf{x}^{(i)}$ and nearly 0 on all others. As a result, the empirical score reduces to $s_\star(\mathbf{z}_t, t) \approx \left[ -z_t + \alpha_t \mathbf{x}^{(i)} \right] / \sigma_t^2$, i.e., a vector pointing to the nearest training data (Figure 1, green arrow). Therefore, despite the elaborate training objective, in fact, the model receives a *trivial* supervision signal inside $\mathcal{T}_t(\delta)$ for most timesteps. We revisit this observation in Section 4 to examine its impact on model generalization.

### 3.2 EXTRAPOLATION DOMINATES DURING INFERENCE

We now turn to inference and empirically justify the claim below:

**C2.** *During inference, diffusion models mostly extrapolate (leave the training region early).*

To examine this, we examine sampling trajectories during inference and show that the sampling trajectory deviates from these shells.

**Experiment 3.3** (Extrapolation During Inference). Recall from Proposition 3.1 that $\mathbf{z}_t \sim \hat{p}_t$ lies in some $i$-th shell, characterized by $\|\mathbf{z}_t - \alpha_t \mathbf{x}^{(i)}\| \approx \sigma_t \sqrt{d}$, with high probability. To measure how much a given $\mathbf{z}_t$ *deviates* from the $\delta$-training region, we define the following quantity:

$$r^{(i)} := \frac{\|\mathbf{z}_t - \alpha_t \mathbf{x}^{(i)}\|}{\sigma_t \sqrt{d}}, \quad r_\star := r^{(i_\star)} \text{ where } i_\star = \underset{i \in \{1, \dots, N\}}{\arg\min} \left| r^{(i)} - 1 \right|. \quad (4)$$

If $r_\star \approx 1$, $\mathbf{z}_t$ lies in the supervision region; otherwise, $\mathbf{z}_t$ deviates from the region. We conduct this experiment on the ImageNet dataset using a pretrained SiT-XL model (Ma et al., 2024). Figure 1b visualizes the distribution of $r_\star$ during both training and inference. As discussed in Section 3.1, $r_\star$ remains tightly concentrated around 1 during training (blue line). In contrast, $r_\star$ rapidly increases above 1 during inference (red line), indicating that the model escapes the supervision region very early at inference ($t = 1 \rightarrow 0$). This demonstrates **C2**: *the model indeed extrapolates during inference, except possibly at the earliest steps.*

# 4 GENERALIZATION THROUGH THE LENS OF SELECTIVE UNDERFITTING

We now characterize our Perspective B using the *supervision region* defined in Section 3.1.

$$\mathbf{s}_{\boldsymbol{\theta}}(\mathbf{z}_t, t) = \begin{cases} \approx \mathbf{s}_\star(\mathbf{z}_t, t) & (\mathbf{z}_t \in \mathcal{T}_t(\delta), \textit{Supervision}) \rightarrow \text{underfitting } \boldsymbol{\times} \\ \not\approx \mathbf{s}_\star(\mathbf{z}_t, t) & (\mathbf{z}_t \notin \mathcal{T}_t(\delta), \textit{Extrapolation}) \rightarrow \text{underfitting } \checkmark \end{cases}$$

During training, supervision occurs only within a set of narrow shells $\mathcal{T}_t(\delta)$, where the empirical score $\mathbf{s}_\star$ reduces to a trivial form (Section 3.1). We claim that underfitting does *not* occur in this region–the model learns toward the empirical score function. At inference time, a sampling trajectory leaves these shells after just a few denoising steps (Section 3.2), and in this extrapolation region, we claim the underfitting *does* occur. In the following Section 4.1, we demonstrate this *selective underfitting* by examining the model's behavior across these two regions.

## 4.1 SELECTIVE UNDERFITTING

**Experiment 4.1** (Supervision vs. Extrapolation). We quantify "underfitting" as the deviation of the learned score from the empirical score, i.e., $\|\mathbf{s}_{\boldsymbol{\theta}} - \mathbf{s}_\star\|^2$. Specifically, we measure this quantity separately within the supervision and extrapolation regions, using pretrained models across various scales: SiT-{S, B, L, XL}. Our experiments are conducted on ImageNet, where the empirical score is computable, facilitating this analysis. Figure 3 reveals that as model capacity increases (left to right), $\mathbf{s}_{\boldsymbol{\theta}}$ approaches $\mathbf{s}_\star$ in the supervision region (blue line), indicating that underfitting does not occur there. Conversely, in the extrapolation region, larger models increasingly deviate from $\mathbf{s}_\star$ (red line), indicating that underfitting occurs. This contrast provides clear evidence for selective underfitting.

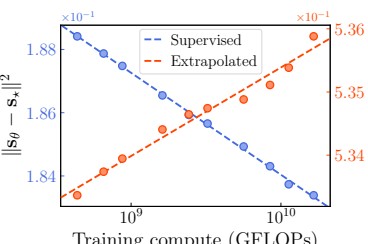

Figure 3: **Quantitative verification of selective underfitting.** $\|\mathbf{s}_{\boldsymbol{\theta}} - \mathbf{s}_\star\|^2$ exhibits contrastive scaling behavior in supervision region vs. extrapolation region.

Experiment 4.1 quantitatively demonstrates that the learned score closely matches the empirical score in the supervision region. But how small is this difference in absolute terms? Below, Experiment 4.2 shows it is small enough for the model to *reproduce* training samples.

**Experiment 4.2** (Memorization in Supervision Region). We take a noisy image $\mathbf{z}_t = \alpha_t \mathbf{x}^{(i)} + \sigma_t \boldsymbol{\epsilon} \sim \hat{p}_t$ (with $\mathbf{x}^{(i)} \in \mathcal{D}$) and use a pretrained SiT-XL model to iteratively denoise from timestep $t$ to 0. Figure 4 shows that for $t < 0.8$, sampling consistently regresses to the original image $\mathbf{x}^{(i)}$. This indicates that the model has (almost perfectly) learned the empirical score in the supervision region, thereby *memorizing* the training image. Notably, previously observed phase transitions (Georgiev et al., 2023; Li & Chen, 2024) can be interpreted as manifestations of selective underfitting.

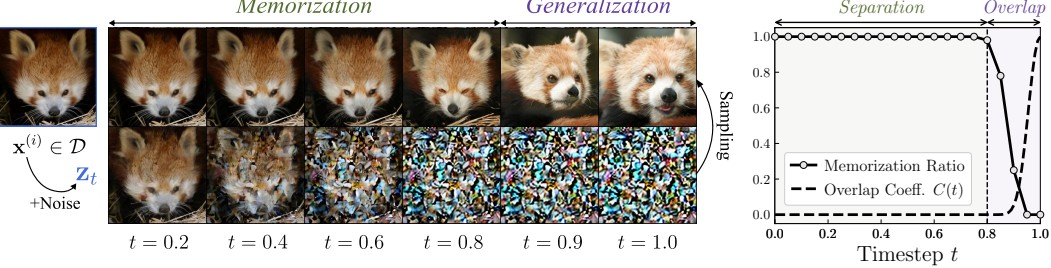

Figure 4: **Qualitative example of selective underfitting.** *Left*: Sampling ($t \rightarrow 0$) from a noisy image $\mathbf{z}_t$ in the supervision region maps back to the original training image $\mathbf{x}^{(i)}$ for most time steps ($t < 0.8$). *Right*: The overlap coefficient $C(t)$ and memorization ratio. The memorization ratio exhibits the same trend quantitatively.

**Selective Underfitting Persists in Gaussian Distribution.** A natural follow-up question is whether this persists even when the ground-truth distribution is simple. Here, the ground truth distribution conceptually denotes the distribution from which the training data are drawn. (For images, this notion is ill-defined and outside our scope). We consider a simple synthetic setup where the ground truth distribution is Gaussian, whose score is linear: in principle, a neural network could fit

this linear function globally, i.e., globally underfit the empirical score, rather than adopting a more complex selective underfitting. Nevertheless, selective underfitting is still observed, suggesting the phenomenon persists even for simple distributions, warranting theoretical investigation of *selective underfitting* (see the result in Appendix C.6).

### 4.2 FREEDOM OF EXTRAPOLATION: A MECHANISM BEHIND GENERALIZATION

We now show how Perspective B offers a novel perspective on diffusion model generalization–one that prior perspectives could not support. The diffusion model generalization refers to a model's ability to generate unseen (creative) data samples. As noted in the introduction, if a model perfectly learns the empirical score function, it would simply regenerate the training set; this is puzzling, since the DSM loss (Equation (1))'s unique minimizer is the empirical score.

The prevailing account, following Perspective A, posits that training-time bias causes the learned score to underfit globally across the data space, leading to a favorable score function. In contrast, we propose that the learned score results from *selective underfitting*–distinct behavior in the supervision versus extrapolation regions. We now show that this distinction is key to generalization by claiming:

**C3.** *Diffusion models fail to generalize when the supervision region is broadened.*

The intuition is that the learned score function is sensitive to the size of the supervision region. Consider two diffusion models trained to minimize $\mathbb{E}_{\mathbf{z}_t \sim q_t} \| \mathbf{s}_{\boldsymbol{\theta}}(\mathbf{z}_t, t) - \mathbf{s}_{\star}(\mathbf{z}_t, t) \|^2$ under different training distributions $q_t$ but with the same $\mathbf{s}_{\star}$. Despite sharing the same empirical score, they can learn markedly different score functions—and hence exhibit different generalization. We refer to this intuition as *Freedom of Extrapolation*: the more *freedom* the model has to extrapolate beyond the supervision region, the better its generalization tends to be.

In standard DSM training (Figure 5a), the network is supervised within a restricted supervision region (blue) while retaining *freedom to extrapolate* (underfit) elsewhere (brown). Conversely, in Figure 5b, enlarging the supervision region (blue) shrinks the space available for extrapolation, yielding *limited freedom to extrapolate* (brown). This behavior cannot be explained by Perspective A, which lacks a notion of a supervision region. We verify this claim with a controlled experiment below.

**Experiment 4.3** (Experimental Verification). We train a series of models that all learn the same empirical score function, but over varying size of supervision regions–specifically, by changing the (effective) support of $\hat{p}_t$ while keeping $s_{\star}$ fixed in the objective $\mathbb{E}_{\mathbf{z}_t \sim \hat{p}_t} \|\mathbf{s}_{\boldsymbol{\theta}}(\mathbf{z}_t, t) - \mathbf{s}_{\star}(\mathbf{z}_t, t)\|^2$. To do this, we select two (random) subsets from the ImageNet training set $\mathcal{D}$: the *score subset* $\mathcal{D}_{\text{score}}$ and the *region subset* $\mathcal{D}_{\text{region}}$, with $\mathcal{D}_{\text{score}} \subseteq \mathcal{D}_{\text{region}} \subseteq \mathcal{D}$. In the training objective, two subsets play distinct roles: $\mathcal{D}_{\text{score}}$ defines the empirical score $s_{\star}$, while $\mathcal{D}_{\text{region}}$ determines the supervision region $\hat{p}_t$. Concretely, we sample $\mathbf{z}_t \sim \hat{p}_t$ by adding noise to points from $\mathcal{D}_{\text{region}}$, and minimize $\mathbb{E}_{\mathbf{z}_t \sim \hat{p}_t} \|\mathbf{s}_{\boldsymbol{\theta}}(\mathbf{z}_t, t) - \mathbf{s}_{\star}(\mathbf{z}_t, t)\|^2$, where $s_{\star}$ is computed with respect to $\mathcal{D}_{\text{score}}$. Consequently, varying only $|\mathcal{D}_{\text{region}}|$ lets us cleanly assess how the supervision-region size affects generalization.

We present the results in Figure 5, fixing $|\mathcal{D}_{\text{score}}|$ and varying $|\mathcal{D}_{\text{region}}|$ from $|\mathcal{D}_{\text{score}}|$ to $|\mathcal{D}|$. Regardless of architecture, when $|\mathcal{D}_{\text{region}}| = |\mathcal{D}_{\text{score}}|$ (*with freedom of extrapolation*), the model *generalizes*; when $|\mathcal{D}_{\text{region}}| \gg |\mathcal{D}_{\text{score}}|$ (*without freedom of extrapolation*), the model *memorizes*, verifying **C3**.

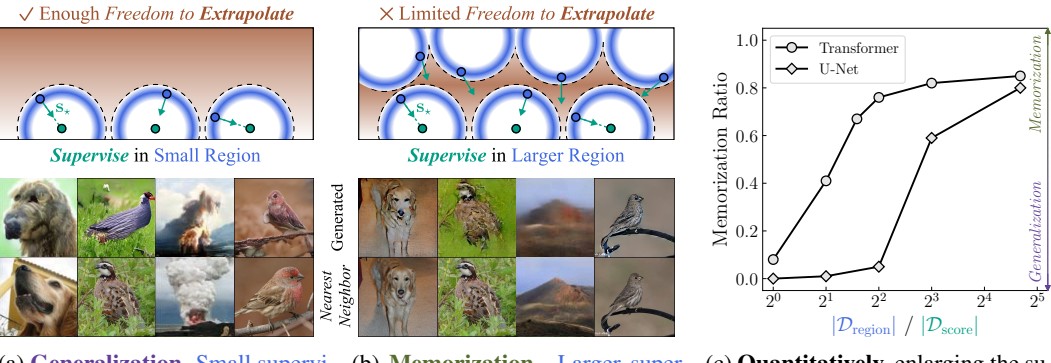

(a) **Generalization.** Small supervision region ($|\mathcal{D}_{\text{region}}| = |\mathcal{D}_{\text{score}}|$) ensures *freedom to extrapolate*.

(b) **Memorization.** Larger supervision region ($|\mathcal{D}_{\text{region}}| \gg |\mathcal{D}_{\text{score}}|$) limits *freedom to extrapolate*.

(c) **Quantitatively,** enlarging the supervision region $\mathcal{D}_{\text{region}}$ increases the ratio of *memorized* samples.

Figure 5: **Experimental verification of *freedom of extrapolation* on ImageNet.** Deliberately limiting the *freedom to extrapolate* transforms the diffusion model from *generalization* to *memorization*.

**Connection to Benign Overfitting.** Benign overfitting (Bartlett et al., 2020; Zou et al., 2021; Frei et al., 2023) in classical supervised learning posits that implicit training biases (e.g., from the optimizer or regularization) guide the learner toward a "favorable" interpolating solution—one that attains zero training loss that generalizes to test data. Our selective underfitting in diffusion models is analogous: the learned score *overfits* the empirical score in the supervision region, while there remain infinitely many extrapolations yielding nearly the same training loss–and *freedom of extrapolation* allows the model to select a favorable score function that generalizes. A difference is that the DSM loss is averaged over the entire data space, so the training loss cannot be zero even if a model fits the empirical score in the supervision region. This analogy opens a path to translating insights from classical supervised learning into a theory of generalization for diffusion models.

**Discussion.** Our claim in **C3** posits that the narrow supervision used in DSM grants diffusion models the *freedom to extrapolate* beyond the supervision region, and that this freedom is central to generalization. Our architecture-agnostic experiments show that this effect persists regardless of inductive bias. **This behavior cannot be explained by previous Perspective A (Kamb & Ganguli, 2024; Niedoba et al., 2024b), which focus on *how* inductive bias shapes underfitting of the empirical score. Our Perspective B adds a complementary dimension: to fully understand diffusion model generalization, one must consider *where* the model underfits, not only *how*.**

## 5 ANALYZING GENERATIVE PERFORMANCE VIA SELECTIVE UNDERFITTING

In this section, we show how our selective underfitting perspective provides a novel viewpoint on *generative performance* of diffusion models–their ability to produce high-quality samples. We first present our findings on REPA (Yu et al., 2024) and, motivated by this, we introduce a quantitative scaling law framework for analyzing generative performance (Section 5.1). Throughout, we use FID (*lower is better*, Heusel et al. (2017)) as the primary metric for measuring generative performance.

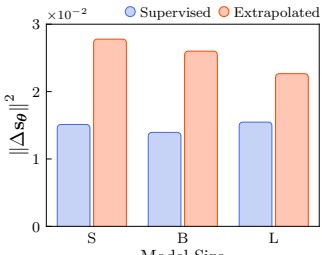

**Our Findings on REPA.** REPA augments the standard DSM loss with an additional regularization term that aligns the model's intermediate representations with features from a pretrained encoder. This simple modification yields a substantial improvement in FID. Interestingly, we find that REPA affects supervision and extrapolation differently. Specifically, we measure the $\ell_2$ difference between the learned scores of SiT models trained with and without REPA–both in the supervision region and the extrapolation region. As shown in Figure 6, REPA *barely alters* the model's behavior in the supervision region, but *greatly alters* its extrapolation behavior. This observation not only clarifies REPA's impact on model extrapolation, but also *motivates* a comprehensive tool to understand generative performance.

Figure 6: **Effect of REPA on the learned score $s_\theta$.** Greater change $\|\Delta s_\theta\|^2$ occurs for extrapolated scores than for supervised scores.

**Motivation.** Recall that the DSM loss in Equation (3) measures how well a model fits the empirical score in the supervision region, serving as a measurement of *supervision*. This supervision loss typically correlates with generative performance: as model size increases, both supervision loss and FID decrease. However, as seen in the REPA case above–where supervision changes were minor despite large FID improvements–*supervision loss alone cannot fully explain FID*. This motivates us to decompose the generative performance into two components: supervision and extrapolation.

### 5.1 DECOMPOSED ANALYSIS OF GENERATIVE PERFORMANCE

We introduce our framework for analyzing generative performance, which can be formulated as

$$\text{Generative Performance} := \text{FID} = f_{\text{extrapolation}}(\text{supervision loss } \mathcal{L}). \quad (5)$$

In words, generative performance is decomposed into two components: (i) supervision loss $\mathcal{L}$, which measures how well the empirical score is fitted by the *model*, and (ii) extrapolation function $f_{\text{extrapolation}}$, which characterizes the extrapolation behavior of the *training recipe*. Conceptually, $f_{\text{extrapolation}}$ acts as a transfer function mapping supervision loss to generative performance. Figure 7a illustrates this: for a fixed training recipe A, represented by $f_{\text{extrapolation}}^{\text{A}}$, smaller supervision loss leads to better FID (*solid line*). Changing the training recipe to B, e.g., modifying the architecture or adding a regularization term, alters the extrapolation function to $f_{\text{extrapolation}}^{\text{B}}$ (*dashed lines*).

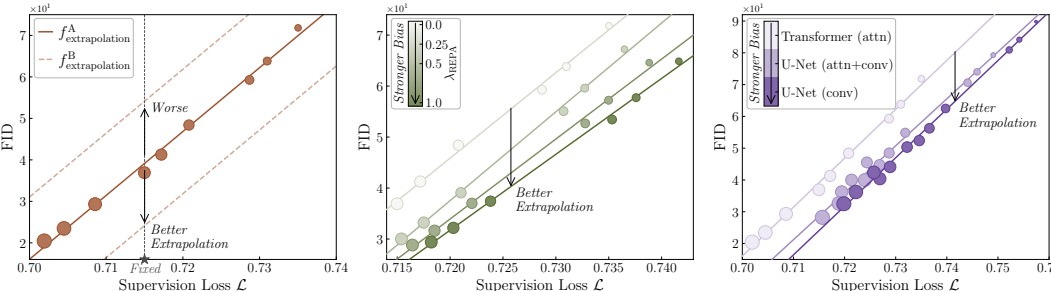

Figure 7: **Illustration of the decomposed analysis of generative performance.** (a) FID ($y$-axis) is plotted as a linear function $f_{\text{extrapolation}}$ (*line*) of supervision loss $\mathcal{L}$ (*x-axis*). $f_{\text{extrapolation}}$ depends on the training recipe (A vs. B) and is more efficient when shifted downward. Our analysis shows that (b) REPA and (c) U-Net (vs. transformers) yield more efficient $f_{\text{extrapolation}}$, i.e., *better extrapolation behavior*. Marker size indicates each model's total training compute (FLOPs).

This framework provides a principled way to assess the efficiency of a *training recipe alone*. A downward shift of the $f_{\text{extrapolation}}$ line indicates greater efficiency–achieving better FID under the same supervision loss–while an upward shift signals reduced efficiency in converting supervision into generative performance. As shown in Figure 7b, consistent with our earlier observations, REPA yields a notably efficient $f_{\text{extrapolation}}$ for diffusion models, explaining much of its empirical success.

**Transformer vs. U-Net.** We next examine the impact of architecture–an important axis of the training recipe–by comparing transformer (attention), U-Net (convolution), and U-Net (attention+convolution) models. As shown in Figure 7c, (i) architectures with more convolutional layers exhibit more efficient extrapolation behavio (downward line shift); but (ii) for a fixed compute budget (marker size), convolution-based architectures have much worse supervision loss than attention-based counterparts. In summary, transformers, compared to U-Nets, yield worse extrapolation efficiency ($\mathcal{L} \rightarrow$ FID), but much scalable supervision efficiency (FLOPs $\rightarrow \mathcal{L}$). This decomposition explains the trend toward transformers for large-scale diffusion training: taking the chain of the two factors, transformers achieve better overall compute efficiency (FLOPs $\rightarrow$ FID). Our results also suggest that one could combine the best of both worlds–the extrapolation efficiency of U-Nets and the supervision efficiency of transformers–as in recent works (Tian et al., 2024; Wang et al., 2024b).

### 5.2 Unified Hypothesis for Better Generative Performance

In Section 5.1, we introduced a framework for quantifying the extrapolation behavior of different training recipes and presented examples that yield more efficient $f_{\text{extrapolation}}$, potentially leading to improved generative performance—a property of practical interest. This naturally raises the question: *is there a simple principle for designing training recipes that induce more efficient extrapolation?* To this end, we propose a unified hypothesis: *Perception-Aligned Training* (PAT).

**PAT.** As shown in Section 3, a diffusion model *extrapolates* beyond the supervision region to generate samples; this geometrically corresponds to *interpolating* among training images. Figure 8 shows two scenarios: perceptually distant images may be close in Euclidean space, so naive interpolation produces perceptually bad samples; when the space is aligned with *perception*, it yields perceptually good samples. This highlights the importance of perception and motivates the following hypothesis:

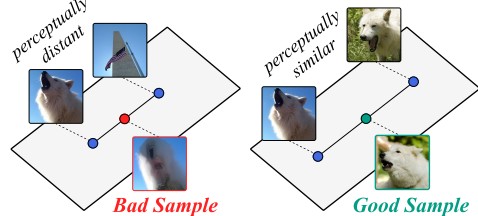

Figure 8: **Good vs. bad samples and their connection to *perception*.**

**C4.** *Aligning a neural network with perception leads to better extrapolation behavior.*

The principle of PAT is illustrated in Figure 9a. Formally, PAT refers to any training recipe that encourages a neural network $\mathbf{s}_{\boldsymbol{\theta}}$ to produce closely aligned outputs $\mathbf{s}_{\boldsymbol{\theta}}(\mathbf{z}_t, t)$ for perceptually close inputs $\mathbf{z}_t$. Below, we present three representative instances of PAT, demonstrating how it unifies many practical training approaches (see Appendix A.3 for further details). We also verify our hypothesis for each instance using 2D toy data (Appendix C.9).

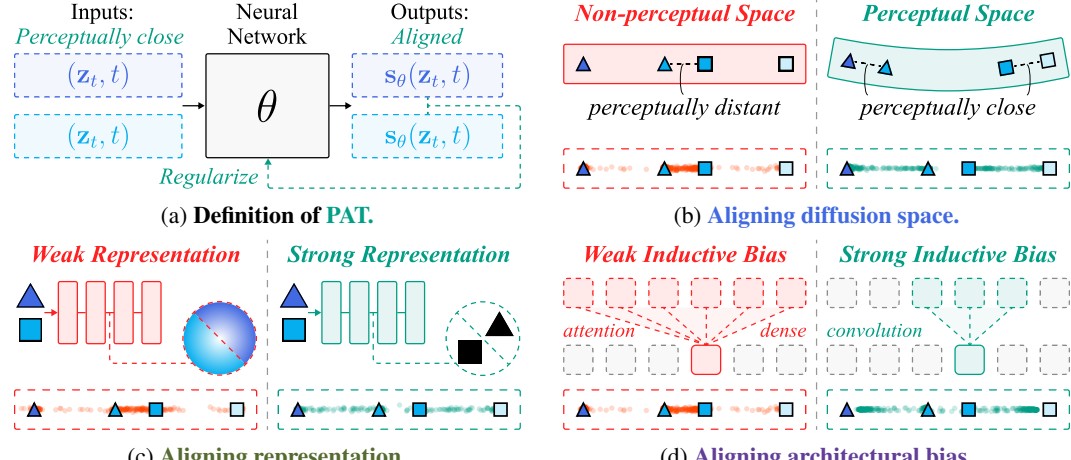

Figure 9: **Illustration of *Perception-Aligned Training* (PAT) with toy experiment results.** PAT regularizes a neural network $\theta$ to output *aligned* scores $\mathbf{s}_\theta$ for *perceptually close* inputs $\mathbf{z}_t$. Many successful training techniques can be unified under this view, categorized as the three instances.

**Aligning Diffusion Space.** The most straightforward instance of PAT is training a model in a perceptually aligned space (Figure 9b). Pixel space and standard VAE latent space, on which diffusion models are commonly trained, often entangle non-perceptual factors (Skorokhodov et al., 2025), making Euclidean distance a poor proxy for perceptual similarity. In contrast, a perceptually aligned space allows a smooth score network $\mathbf{s}_\theta$ to naturally produce consistent outputs for perceptually similar inputs. This perspective explains the success of methods that construct perceptually meaningful latent spaces, such as EQ-VAE (Kouzelis et al., 2025a) and VA-VAE (Yao et al., 2025).

**Aligning Representation.** Another approach aligns the model's internal representation with perception (Figure 9c). While keeping output supervision unchanged, the training objective encourages perceptually meaningful intermediate features, which implicitly regularize the outputs. Empirically, this yields more favorable extrapolation (Figure 7b). Methods incorporating representation learning into diffusion training–REPA (Yu et al., 2024) and MaskDiT (Zheng et al., 2023)–are examples.

**Aligning Architectural Bias.** A third instance is aligning the network's inductive bias with perception (Figure 9d). For images, convolution imparts translational equivariance and locality, aligning well with perceptual invariance such as translation or cropping. This explains the reason why convolution-based architectures tend to induce a more efficient extrapolation function $f_{\text{extrapolation}}$ (Figure 7c). However, as noted in Section 5.1, these designs are less favorable in supervision efficiency (FLOPs $\rightarrow \mathcal{L}$), which often limits their practical effectiveness at large training scales.

**Discussion.** We posit that a diffusion model's training efficiency decomposes into **(i)** *supervision efficiency*: how well the model fits the training data, and **(ii)** *extrapolation efficiency*: how this fit translates into generation performance. This decomposition enables a thorough analysis of different *training recipes*, beyond simply comparing per-model FID. With this perspective, PAT offers a unified principle for designing training methods that induce a favorable extrapolation function, i.e., improving **(ii)**. However, PAT can sometimes weaken **(i)**; thus, maximizing generative performance under a fixed compute requires balancing supervision and extrapolation. Finally, we emphasize that **these practical tools are a natural byproduct of Perspective B, impossible with Perspective A**.

## 6 CONCLUSION

In this work, we argue that diffusion models exhibit distinct behaviors in supervision and extrapolation regions–a perspective we call *selective underfitting*, which offers a novel viewpoint on their generalization and generative performance. Theoretically, however, the mechanism enabling extrapolation remains unclear; future work may further analyze *how the score in the extrapolation region is shaped*, building on our findings about the freedom of extrapolation. Practically, while we do not propose a new training method that optimally balances supervision and extrapolation, our results suggest that *many well-balanced training algorithms can be developed using our decomposed analysis framework*. Ultimately, we expect these insights to extend to other generative models, contributing to a deeper understanding and the development of more powerful generative systems.

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

CONTENTS

# A DISCUSSION

## A.1 RELATED WORK

**Empirical Score of Diffusion Models.** The primary formulations of diffusion models (Ho et al., 2020; Song et al., 2020; Sohl-Dickstein et al., 2015) did not explicitly present the analytic form of the empirical score function, and it took several years for the community to recognize that standard training learns precisely this quantity, yielding the expression in Equation (2). This realization has sparked several lines of work: the mainstream thread seeks to understand diffusion model generalization, which we review comprehensively review in the next paragraph. In parallel, the analytic form has been used to study the geometry of ODE sampling trajectories (Chen et al., 2025; Wan et al., 2024) and to improve diffusion model training in various ways (Niedoba et al., 2024a; Xu et al., 2023; Scarvelis et al., 2023; Xu et al., 2025).

**Diffusion Model Generalization.** Early works (Gu et al., 2023; Yoon et al., 2023) showed a transition from memorization to generalization as the training dataset size grows under fixed model size. Moreover, if a model perfectly learns the empirical score–i.e., if the DSM loss in Equation (3) is fully minimized–it exactly regenerates the training data. This principle-practice discrepancy has motivated efforts to understand diffusion model generalization. Several works argue and empirically show that, while the model is supervised with the empirical score, inductive biases lead it to *globally underfits* the empirical score, resulting in learning a favorable score function. These bias includes architectural aspects (Kamb & Ganguli, 2024; Kadkhodaie et al., 2023; Niedoba et al., 2024b), sampling and optimization effects (Yi et al., 2023; Wu et al., 2025b), and stochasticity (Vastola, 2025). Compared to these works that focus on how a model *globally underfits* the empirical score, we divide the data space into two regions and propose *selective underfitting*–where underfitting occurs only in the extrapolation region–as a complementary viewpoint to understand generalization.

Other studies have found systematic differences between the learned and empirical scores. Wang & Vastola (2024); Li et al. (2024) find that the learned score exhibits an approximately linear structure, while Chen (2025); Wang et al. (2024a) show that it is substantially smoother than the empirical score. Complementing these results, Bertrand et al. (2025) argues that DSM stochasticity is unlikely to be the primary driver of generalization, and Zhang et al. (2025) proposes a generalization metric that yields accurate scaling laws. Theoretically, Farghly et al. (2025) uses algorithmic stability to argue that these models can generalize toward the ground-truth score. In contrast, Shah et al. (2025) studies Ambient Diffusion–a variant of vanilla DSM trained on noisy images–and suggests that generalization can occur without a selective-underfitting mechanism. In Ambient Diffusion, the minimizer of its training loss differs from the empirical score, enabling generalization without memorization (in our terms, without fitting the empirical score in the supervision region).

## A.2 CONNECTION TO GENERALIZATION IN SUPERVISED LEARNING

Generalization is a classic problem in deep learning, typically studied in supervised learning, where a model is trained on finite data but is expected to learn a function that goes beyond mere interpolation. Our *selective underfitting* framework differs from this standard generalization studies in two key ways: (1) *training distribution $\neq$ test distribution*: whereas supervised learning assumes train and test data from the same distribution, diffusion models operate with distinct distributions, which we verify in Section 3. (2) *existence of the empirical score*: diffusion models admit a natural, analytically defined empirical score that is well-defined on the entire data space. This empirical score serves as a canonical interpolant of the 'overfitted' solution, thereby enabling more precise analysis of generalization. In contrast, standard supervised learning lacks such a canonical choice.

We note a structural connection between our selective underfitting and benign overfitting (as we mention in Section 4.1). As noted in the abstract of Yoon et al. (2023), unlike benign overfitting, a diffusion model that perfectly fits the empirical score on the entire data space achieves identically zero training loss, which precludes generalization. Our perspective comes from a less tight viewpoint; since the training distribution is highly concentrated in the supervision region, a diffusion model that fits the empirical score there but remains to extrapolate beyond achieves a nearly zero training loss. Therefore, our analogy is structural (overfit-inside / free-outside), not literal (zero-loss benign overfitting).

## A.3 Examples of Perception-Aligned Training Recipes

In Section 5.2, we argued that many practical training techniques can be unified under the Perception-Aligned Training (PAT) framework. Here, we discuss specific works interpreted as PAT, grouped into three main instances.

**Aligning Diffusion Space.** Many recent works have reported significant performance improvements by constructing perceptually aligned latent spaces, albeit through different strategies: enforcing spatial equivariance (EQ-VAE, Kouzelis et al., 2025a), aligning with a vision foundation model (VA-VAE, Yao et al., 2025; REPA-E, Leng et al., 2025), concatenating semantic features from a vision foundation model to form a new perceptual space (ReDi, Kouzelis et al., 2025b), or regularizing to suppress non-perceptual information (Skorokhodov et al., 2025). Given these successes, which are grounded in the PAT principle, we expect that building more perceptually aligned autoencoders will further improve the training efficiency of diffusion models.

**Aligning Representation.** Incorporating representation learning into diffusion training has proven highly effective. The pioneering work REPA (Yu et al., 2024) aligns the model's internal representation with perceptual features from a pretrained encoder such as DINO (Oquab et al., 2023), resulting in significant performance gains. Follow-up works have explored representation learning without external encoders, using self-distillation (Jiang et al., 2025) or contrastive losses (Wang & He, 2025). Alternatively, methods like MDT (Gao et al., 2023) and MaskDiT (Zheng et al., 2023) directly incorporate masked reconstruction objectives, which are known to promote strong internal representations. Despite the variety of strategies, these approaches all share the common mechanism of improving extrapolation by enhancing the model's internal representation.

**Aligning Architectural Bias.** Transformer-based architectures (with *weaker* inductive bias), such as DiT (Peebles & Xie, 2023) and SiT (Ma et al., 2024), have become standard, outperforming convolution-based architectures (with *stronger* inductive bias), such as U-Net (Ronneberger et al., 2015; Ho et al., 2020). As discussed in Section 5.1, this is because convolutional models have superior extrapolation efficiency but much worse supervision efficiency. Interestingly, recent works have managed to balance these two efficiency factors and surpass the performance of pure diffusion transformers, including hybrid architectures that incorporate convolution layers into transformers (U-DiT, Tian et al., 2024; Swin-DiT, Wu et al., 2025a), as well as purely convolutional models carefully tuned for efficiency and scalablity (FlowDCN, Wang et al., 2024b; DiC, Tian et al., 2025). This highlights the importance of balancing extrapolation and supervision efficiency to enable compute-efficient diffusion training.

# B Theoretical Details

## B.1 Concentration of the Supervision Region

**Proposition 3.1** (Supervision region). *Let $\delta \in (0,1)$, empirical data $\{\mathbf{x}^{(i)}\}_{i=1}^{N}$, and $\mathbf{z}_t \sim \hat{p}_t$. Then*

$$\mathbb{P}(\mathbf{z}_t \in \mathcal{T}_t(\delta)) \geq 1 - \delta, \quad \mathcal{T}_t(\delta) \colon = \left\{ \mathbf{z} : \exists i \in [N], \left| \|\mathbf{z} - \alpha_t \mathbf{x}^{(i)}\| - \sigma_t \sqrt{d} \right| \leq \sigma_t \sqrt{d \log(1/\delta)} \right\}.$$

*Proof.* We use the well-known fact that for a $d$-dimensional standard Gaussian vector $\boldsymbol{\epsilon} \sim \mathcal{N}(0, \boldsymbol{I})$,

$$\left| \|\boldsymbol{\epsilon}\| - \sqrt{d} \right| \leq \sqrt{d \log(1/\delta)}$$

which holds with probability at least $1 - \delta$. Next, recall the $\delta$-supervision region

$$\mathcal{T}_t(\delta) \colon = \left\{ \mathbf{z} : \exists i \in [N], \left| \|\mathbf{z} - \alpha_t \mathbf{x}^{(i)}\| - \sigma_t \sqrt{d} \right| \leq \sigma_t \sqrt{d \log(1/\delta)} \right\}.$$

Therefore, for $\mathbf{z}_t \sim \hat{p}_t$,

$$\mathbb{P}\left(\mathbf{z} \notin \mathcal{T}_t(\delta)\right) \leq \frac{1}{N} \sum_{i=1}^{N} \mathbb{P}_{\boldsymbol{\epsilon} \sim \mathcal{N}(0,\boldsymbol{I}), \mathbf{z} = \alpha_t \mathbf{x}^{(i)} + \sigma_t \boldsymbol{\epsilon}} \left( \left| \|\mathbf{z} - \alpha_t \mathbf{x}^{(i)}\| - \sigma_t \sqrt{d} \right| > \sigma_t \sqrt{d \log(1/\delta)} \right) \leq \delta.$$

$\square$

### B.2 Non-Overlapping Property of the Supervision Region

We quantify distributional overlap of supervised shells using the Bhattacharyya coefficient: for two distributions $p$ and $q$ on $\mathbb{R}^d$,

$$0 \leq \int_{\mathbb{R}^d} \sqrt{p(\mathbf{z})q(\mathbf{z})} \mathrm{d}\mathbf{z}_t \leq 1.$$

For the Gaussians $\mathcal{N}(\mathbf{z}_t; \alpha_t \mathbf{x}^{(i)}, \sigma_t \boldsymbol{I})$, the worst-case Bhattacharyya coefficient across shells simplifies to

$$C(t) := \max_{i \neq j} \int_{\mathbb{R}^d} \sqrt{\hat{p}_t^{(i)}(\mathbf{z}_t) \hat{p}_t^{(j)}(\mathbf{z}_t)} \mathrm{d}\mathbf{z}_t = \max_{i \neq j} \left[ \exp\left( -\frac{\alpha_t^2 \left\| \mathbf{x}^{(i)} - \mathbf{x}^{(j)} \right\|^2}{8\sigma_t^2} \right) \right].$$

Evaluating this constant on the ImageNet dataset (Deng et al., 2009) and in the latent space of the SD-VAE (Rombach et al., 2022) yields Figure 4.

## C  Experimental Details

### C.1  Default Experimental Setup

As many of our experiments share a similar setup, we begin by outlining the default configuration. Most implementation details and hyperparameters follow those of SiT (Ma et al., 2024).

**Dataset.** We conduct all of our real-world image generation experiments on ImageNet at $256 \times 256$, consisting of $N = 1281167$ images across 1000 classes. We apply horizontal flip augmentation during preprocessing, effectively doubling the dataset size to $|\mathcal{D}| = 2N$.

**Diffusion.** We use simple linear interpolants with $\alpha_t = 1 - t$ and $\sigma_t = t$, and adopt the $\mathbf{v}$-prediction objective. The velocity $\mathbf{v}_{\boldsymbol{\theta}}$ is related to the score $\mathbf{s}_{\boldsymbol{\theta}}$ by the linear relation $\mathbf{v}_{\boldsymbol{\theta}}(\mathbf{z}_t, t) = -\frac{t}{1-t}\mathbf{s}_{\boldsymbol{\theta}}(\mathbf{z}_t, t) - \frac{1}{1-t}\mathbf{z}_t$. As an exception, models in Experiment 4.3 uses $\mathbf{x}$-prediction for numerical stability, which is also linearly related to the score: $\mathbf{x}_{\boldsymbol{\theta}}(\mathbf{z}_t, t) = \frac{t^2}{1-t}\mathbf{s}_{\boldsymbol{\theta}}(\mathbf{z}_t, t) + \frac{1}{1-t}\mathbf{z}_t$. For sampling, we use the `dopri5` adaptive ODE solver with `atol=1e-6` and `rtol=1e-3`.

**Training.** Models are trained on 4 NVIDIA H100 GPUs with the AdamW optimizer (Loshchilov & Hutter, 2017) (no weight decay), with $(\beta_1, \beta_2) = (0.9, 0.999)$, a constant learning rate of $10^{-4}$, and batch size of 256 for 400K training iterations. We maintain an exponential moving average (EMA) of model weights with a decay rate of 0.9999. All models are latent diffusion models trained on $32 \times 32 \times 4$ latents precomputed using SD-VAE (Rombach et al., 2022). Additionally, all models are *class-conditional*, modeling 1000 distinct distributions, and trained with a class dropout probability of 0.1 to enable estimation of *unconditional* scores. We employ two different architectures:

(1) Diffusion Transformer: Following the SiT architecture (Ma et al., 2024), with four configurations (S, B, L, XL), using a patch size of 2.

(2) U-Net: Based on Dhariwal & Nichol (2021), with four configurations (XS, S, B, L) varying model channels (Karras et al., 2024) as $(64, 128, 192, 320)$. Attention layers are applied at resolutions $(1, 2, 4)$, except in the convolution-only U-Net, where attention is disabled.

Unless otherwise specified, we primarily use diffusion transformers (SiT) in our experiments.

**Evaluation.** For FID evaluation, we follow the protocol of Dhariwal & Nichol (2021) using their official implementation. FLOPs are measured with `ptflops` (Sovrasov, 2018), and total training compute is calculated as FLOPs per forward pass multiplied by the number of training iterations.

**Supervision and Extrapolation Regions.** Many of our experiments measure the expectation of a *quantity*, $\mathbb{E}[\mathcal{Q}]$, separately over the supervision and extrapolation regions. As these regions correspond to continuous probability distributions in data space, we report Monte Carlo estimates by sampling from each region and averaging the computed quantities.

For the supervision region, we sample $\mathbf{z}_t^{\mathrm{sup}}$ from $\hat{p}_t$ by randomly selecting a training data point $\mathbf{x}$ and Gaussian noise $\boldsymbol{\epsilon}$, then running the forward process: $\mathbf{z}_t^{\mathrm{sup}} = \alpha_t \mathbf{x} + \sigma_t \boldsymbol{\epsilon}$, identical to the training procedure. For the extrapolation region, we sample $\mathbf{z}_t^{\mathrm{ext}}$ from an inference trajectory, obtained by solving the ODE with a pretrained model starting from random noise $\mathbf{z}_1^{\mathrm{ext}} = \boldsymbol{\epsilon}$. We then average the

quantity $\mathcal{Q}$ over $N$ samples:

$$\hat{\mathbb{E}}_t^{\mathrm{sup}}(\mathcal{Q}) = \frac{1}{N} \sum_{i=1}^{N} \mathcal{Q}(\mathbf{z}_t^{i,\mathrm{sup}}, t), \ \ \hat{\mathbb{E}}_t^{\mathrm{ext}}(\mathcal{Q}) = \frac{1}{N} \sum_{i=1}^{N} \mathcal{Q}(\mathbf{z}_t^{i,\mathrm{ext}}, t)$$

Often, we measure $\mathcal{Q}$ across all timesteps rather than per timestep. In these cases, we use a timestep-dependent weighting function $\lambda(t)$ to ensure comparable scales across timesteps. Specifically, we sample $T$ random timesteps $\{t_j\}_{j=1}^{N}$ from $\mathrm{Unif}(0,1)$ and repeat the above process, yielding:

$$\hat{\mathbb{E}}^{\mathrm{sup}}(\mathcal{Q}, \lambda) = \frac{1}{NT} \sum_{i=1}^{N} \sum_{i=1}^{T} \lambda(t_j) \mathcal{Q}(\mathbf{z}_{t_j}^{i,\mathrm{sup}}, t_j), \ \ \hat{\mathbb{E}}^{\mathrm{ext}}(\mathcal{Q}, \lambda) = \frac{1}{NT} \sum_{i=1}^{N} \sum_{i=1}^{T} \lambda(t_j) \mathcal{Q}(\mathbf{z}_{t_j}^{i,\mathrm{ext}}, t_j)$$

Here, $\{\mathbf{z}_{t_j}^{i,\mathrm{sup}}\}_{j=1}^{T}$ vary only in timesteps for the same data-noise pair $(\mathbf{x}, \boldsymbol{\epsilon})$, and $\{\mathbf{z}_{t_j}^{i,\mathrm{ext}}\}_{j=1}^{T}$ vary only in timestep along the same inference trajectory. For the rest of the appendix, we simply specify $N$ and $\mathcal{Q}$ (and $T$ and $\lambda$, if integrating over all timesteps), using the above notations.

## C.2 Details on CFG Paradox Experiment (Example 3.1)

In Example 3.1, we plot the difference between conditional and unconditional scores, $\|\mathbf{s}^c - \mathbf{s}^{\varnothing}\|$, in both the supervision and extrapolation regions. The difference is measured separately at 100 different timesteps, $t = \{0.01, 0.02, \ldots, 1.00\}$, using $\hat{\mathbb{E}}_t^{\mathrm{sup}}(\mathcal{Q})$ and $\hat{\mathbb{E}}_t^{\mathrm{exp}}(\mathcal{Q})$ of Appendix C.1 as described in Appendix C.1. For the supervision region, we use $\mathcal{Q}(\mathbf{z}_t, t) = \|\mathbf{s}_\star^c(\mathbf{z}_t, t) - \mathbf{s}_\star^{\varnothing}(\mathbf{z}_t, t)\|$ (difference of the empirical score), and for the extrapolation region, $\mathcal{Q}(\mathbf{z}_t, t) = \|\mathbf{s}_{\boldsymbol{\theta}}^c(\mathbf{z}_t, t) - \mathbf{s}_{\boldsymbol{\theta}}^{\varnothing}(\mathbf{z}_t, t)\|$ (difference of the learned score) for the extrapolation region, with $N = 200$ samples per timestep. For better visualization, we plot the median (dashed line) and the 10–90% percentile range (shaded area) for each timestep.

## C.3 Details on Extrapolation During Inference Experiment (Experiment 3.3)

In Experiment 3.3, we plot the $r_\star$ values defined in Equation (4) measured in the supervision and extrapolation regions (Figure 1b). Similarly to Appendix C.2, we measure the $r_\star$ values on 100 different timesteps, with $N = 100$ samples per timestep. For clarity, we plot each training data point and each sampling trajectory as a separate line, rather than showing averages.

## C.4 Details on Memorization Experiment (Experiment 4.2)

In Experiment 4.2, we sample $N = 200$ images $\mathbf{x}^{(i)} \in \mathcal{D}$ from the training dataset. For each image, we construct a noisy image $\mathbf{z}_t = \alpha_t \mathbf{x}^{(i)} + \sigma_t \boldsymbol{\epsilon}$ at 21 timesteps $t = 0.00, 0.05, 0.10, \ldots, 1.00$. We then run ODE sampling ($t \to 0$) from each noisy image $\mathbf{z}_t$ and compare the final output to the original image $\mathbf{x}^{(i)}$. Figure 4 (left) shows a qualitative example, while Figure 4 (right) plots the memorization ratio, defined as the fraction of sampled images for which the $\ell_2$-closest image in the training set is the original $\mathbf{x}^{(i)}$ (following the calibrated $\ell_2$ metric in Carlini et al. (2023)). Additionally, the overlap coefficient from Appendix B.2 is measured on the class-conditional distribution for a single class (red panda, id $= 387$).

## C.5 Details on Supervision vs. Extrapolation Experiment (Experiment 4.1)

In Experiment 4.1, we measure the score error $\|\mathbf{s}_\theta - \mathbf{s}_\star\|^2$ in the supervision and extrapolation regions across different model sizes. The score error in each region is computed as $\hat{\mathbb{E}}^{\mathrm{sup}}(\mathcal{Q}, \lambda)$ and $\hat{\mathbb{E}}^{\mathrm{ext}}(\mathcal{Q}, \lambda)$ described in Appendix C.1, using the following parameters: $\mathcal{Q}(\mathbf{z}_t, t) = \|\mathbf{s}_{\boldsymbol{\theta}}(\mathbf{z}_t, t) - \mathbf{s}_\star(\mathbf{z}_t, t)\|^2$, $\lambda(t) = \frac{t^2}{(1-t)^2}$, $N = 1000$, and $T = 100$. The weighting function $\lambda(t)$ is chosen so that $\lambda(t)\mathcal{Q}(\mathbf{z}_t, t) = \|\mathbf{v}_{\boldsymbol{\theta}}(\mathbf{z}_t, t) - \mathbf{v}_\star(\mathbf{z}_t, t)\|^2$, which reflects the $\mathbf{v}$-prediction objective and ensures uniform aggregation of the score error across timesteps.

C.6 VERIFICATION OF SELECTIVE UNDERFITTING IN GAUSSIAN (SECTION 4.1)

In this section, we empirically show that selective underfitting persists even with a Gaussian distribution.

**Setup.** We consider a synthetic setup where the ground truth data distribution is Gaussian: $p_{\text{data}} = \mathcal{N}(0, \boldsymbol{I})$ in $d = 20$ dimensions, with ground truth score $\mathbf{s}_{\mathbf{gt}}(\mathbf{z}_t, t) := -\mathbf{z}_t/\sigma_t$. We construct a finite training set $\mathcal{D}$ by randomly sampling $|\mathcal{D}| = 100$ data points from $p_{\text{data}} = \mathcal{N}(0, \boldsymbol{I})$. We train an MLP (10 layers, width 1000) for 20,000 iterations using Adam (Kingma & Ba, 2014) with learning rate $4 \times 10^{-3}$. Every 100 iterations, we evaluate the learned score $\mathbf{s}_{\boldsymbol{\theta}}$, the empirical score $\mathbf{s}_\star$, and the ground truth score $\mathbf{s}_{\mathbf{gt}}$.

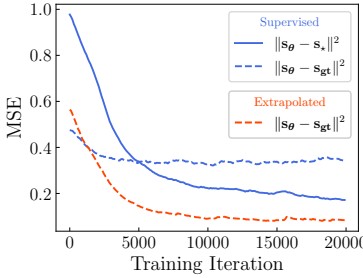

**Result.** Figure 10 visualizes $\|\mathbf{s}_{\boldsymbol{\theta}} - \mathbf{s}_\star\|^2$ and $\|\mathbf{s}_{\boldsymbol{\theta}} - \mathbf{s}_{\mathbf{gt}}\|^2$ in the supervision region (integrated by $\hat{p}_t$), and $\|\mathbf{s}_{\boldsymbol{\theta}} - \mathbf{s}_{\mathbf{gt}}\|^2$ in the entire data space (integrated by $p_{\text{data}}$). In the supervision region, the error between the learned score and the ground truth score plateaus (solid blue), while the learned score converges to the empirical score (dashed blue), indicating no underfitting. In contrast, outside of the supervision region, the model's learned score approaches the ground truth score (dashed red), indicating underfitting. This verifies that selective underfitting persists even for a simple Gaussian.

Figure 10: **Gaussian Verification**.

C.7 DETAILS ON FREEDOM OF EXTRAPOLATION EXPERIMENT (EXPERIMENT 4.3)

We detail the setup of the *freedom of extrapolation* experiment, including subset construction, training procedure, and evaluation metric.

**Score and Region Subsets.** As described in Section 4.2, we use two nested subsets $\mathcal{D}_{\text{score}} \subseteq \mathcal{D}_{\text{region}}$, drawn from the ImageNet training set $\mathcal{D}$. These subsets play distinct roles in the DSM objective: $\mathcal{D}_{\text{score}}$ defines the target (empirical) score $\mathbf{s}_\star$, while $\mathcal{D}_{\text{region}}$ defines the data distribution $\hat{p}_t$ for sampling inputs in the DSM objective:

$$\mathcal{L}_{\text{DSM}}(t) = \mathbb{E}_{\mathbf{z}_t \sim \hat{p}_t} \|\mathbf{s}_{\boldsymbol{\theta}}(\mathbf{z}_t, t) - \mathbf{s}_\star(\mathbf{z}_t, t)\|^2. \tag{6}$$

To construct $\mathcal{D}_{\text{score}}$, we sample 100 images per ImageNet class, resulting in $|\mathcal{D}_{\text{score}}| = 1000 * 100 = 10^5$ images. To obtain $\mathcal{D}_{\text{region}}$ of varying sizes while preserving $\mathcal{D}_{\text{score}}$, we augment $\mathcal{D}_{\text{score}}$ with additional images sampled uniformly without replacement from the remaining images in each class. Fixing $\mathcal{D}_{\text{score}}$ and varying $\mathcal{D}_{\text{region}}$, we train both SiT-L and UNet-B models.

**Training (Importance Sampling).** Directly applying the DSM loss is infeasible when $\mathcal{D}_{\text{score}} \neq \mathcal{D}_{\text{region}}$, since it would require computing the empirical score at every iteration. To address this, we use an importance-sampling technique inspired by Niedoba et al. (2024a). For a given $\mathbf{z}_t$, we define an auxiliary distribution over $\mathcal{D}_{\text{score}} := \{\mathbf{x}^{(i)}\}_{i=1}^n$:

$$r_{\mathbf{z}_t}(\mathbf{y}) \propto \exp\left(-\frac{\|\mathbf{z}_t - \alpha_t \mathbf{y}\|^2}{2\sigma_t^2}\right) : = \text{softmax}\left(-\frac{\|\mathbf{z}_t - \alpha_t \mathbf{y}\|^2}{2\sigma_t^2}\right),$$

Using $r_{\mathbf{z}_t}$, we define the alternative loss:

$$\mathcal{L}'_\theta(t) : = \mathbb{E}_{t, \mathbf{x}, \mathbf{z}_t, \mathbf{y}} \left\| \mathbf{s}_{\boldsymbol{\theta}}(\mathbf{z}_t, t) + \frac{\mathbf{z}_t - \alpha_t \mathbf{y}}{\sigma_t^2} \right\|^2, \quad \mathbf{x} \sim \text{Unif}(\mathcal{D}_{\text{region}}), \ \mathbf{z}_t = \alpha_t \mathbf{x} + \sigma_t \boldsymbol{\epsilon}, \ \mathbf{y} \sim r_{\mathbf{z}_t}. \tag{7}$$

In words, we sample $\mathbf{x}$ from $\mathcal{D}_{\text{region}}$ and form $\mathbf{z}_t$ as in standard DSM training, then additionally sample $\mathbf{y}$ from $r_{\mathbf{z}_t}$.

*Equivalence Proof.* We now show that Equation (6) and Equation (7) are equivalent up to a constant:

$$\mathbb{E}_{t,\mathbf{x},\mathbf{z}_t,\mathbf{y}}\left\|\mathbf{s}_{\boldsymbol{\theta}}(\mathbf{z}_t,t)+\frac{\mathbf{z}_t-\alpha_t\mathbf{y}}{\sigma_t^2}\right\|^2$$

$$=\mathbb{E}_{t,\mathbf{x},\mathbf{z}_t}\left[\int_{\mathbf{y}}\left\|\mathbf{s}_{\boldsymbol{\theta}}(\mathbf{z}_t,t)+\frac{\mathbf{z}_t-\alpha_t\mathbf{y}}{\sigma_t^2}\right\|^2 r_{\mathbf{z}_t}(\mathbf{y})d\mathbf{y}\right]$$

$$=\mathbb{E}_{t,\mathbf{x},\mathbf{z}_r}\left[\left\|\mathbf{s}_{\boldsymbol{\theta}}(\mathbf{z}_t,t)+\int_{\mathbf{y}}\frac{\mathbf{z}_t-\alpha_t\mathbf{y}}{\sigma_t^2}r_{\mathbf{z}_t}(\mathbf{y})\right\|^2 d\mathbf{y}\right]+\mathcal{C}$$

$$=\mathbb{E}_{t,\mathbf{x},\mathbf{z}_t}\left[\left\|\mathbf{s}_{\boldsymbol{\theta}}(\mathbf{z}_t,t)+\sum_{\mathbf{y}\in\mathcal{D}_{\text{score}}}\frac{\mathbf{z}_t-\alpha_t\mathbf{y}}{\sigma_t^2}\text{softmax}\left(-\frac{\|\mathbf{z}_t-\alpha_t\mathbf{y}\|^2}{2\sigma_t^2}\right)\right\|^2\right]+\mathcal{C}$$

$$=\mathbb{E}_{t,\mathbf{x},\mathbf{z}_t}\left[\|\mathbf{s}_{\boldsymbol{\theta}}(\mathbf{z}_t,t)-\mathbf{s}_{\star}(\mathbf{z}_t,t)\|^2\right]+\mathcal{C}.$$

where $\mathcal{C}$ is a constant independent of $\mathbf{s}_{\boldsymbol{\theta}}$. The second equality applies a variance decomposition over $\mathbf{y}\sim r_{\mathbf{z}_t}$ (the resulting variance term does not depend on $\mathbf{s}_{\boldsymbol{\theta}}$ and is absorbed into $\mathcal{C}$), and the last equality uses the definition of the empirical score $\mathbf{s}_{\star}(\mathbf{z}_t,t)$ in Equation (2). $\square$

In summary, this loss is equivalent to the original DSM loss up to a constant, but avoids explicit computation of the empirical score at every iteration, enabling efficient training when $\mathcal{D}_{\text{score}}\neq\mathcal{D}_{\text{region}}$. We implement $r_{\mathbf{z}_t}$ using GPU L2 indexing with Faiss (Johnson et al., 2019; Douze et al., 2024), incurring minimal overhead.

**Evaluation (Memorization Ratio).** After training, we evaluate each model's generalization beyond the score subset $\mathcal{D}_{\text{score}}$ using the *calibrated $\ell_2$ metric* from Carlini et al. (2023). For a generated image $\mathbf{x}$, we compute

$$\frac{\|\mathbf{x}-\mathbf{x}^{'(1)}\|_2^2}{\frac{1}{n}\sum_{i=1}^{n}\|\mathbf{x}-\mathbf{x}^{'(i)}\|_2^2},$$

where $\mathbf{x}^{'(i)}$ is the $i$-th nearest training image in $\mathcal{D}_{\text{score}}$. A value near 0 indicates memorization (the sample is unusually close to a training image); a value near 1 indicates no memorization. We set $n=50$ and, for each model, generate 256 images to compute the memorization ratio.

## C.8    Details on Decomposed Analysis of Generative Performance (Section 5.1)

In this section, we provide details for our decomposed analysis of generative performance.

**Supervision Loss.** The supervision loss $\mathcal{L}$ quantifies how well the model $\mathbf{s}_{\boldsymbol{\theta}}$ approximates the empirical score $\mathbf{s}_{\star}$ within the supervision region. It is computed as $\hat{\mathbb{E}}^{\text{sup}}(\mathcal{Q},\lambda)$ described in Appendix C.1, where $\mathcal{Q}(\mathbf{z}_t,t)=\|\mathbf{s}_{\boldsymbol{\theta}}(\mathbf{z}_t,t)-\mathbf{s}_{\star}(\mathbf{z}_t,t)\|^2$, $\lambda(t)=\frac{t^2}{(1-t)^2}$, $N=1000$, and $T=100$. These are identical parameters to those used to compute score errors in Experiment 4.1.

**REPA.** We closely follow the original implementation (Yu et al., 2024) to incorporate REPA into SiT. Specifically, we align the intermediate representation of SiT at depth 8 with features from DINOv2-B (Oquab et al., 2023) using negative cosine similarity loss weighted by $\lambda_{\text{REPA}}$. To study the effect of stronger representation alignment on the extrapolation function, we vary $\lambda_{\text{REPA}}$ in $\{0.0, 0.25, 0.5, 1.0\}$. For each $\lambda_{\text{REPA}}$ (each line in the scaling plot Figure 9c), each data point corresponds to a model from SiT-$\{S, B\}$ trained for $\{200K, 300K, 400K\}$ iterations.

**Transformer vs. U-Net.** We compare three architectures: Transformer (attention), U-Net (attention+convolution), and U-Net (convolution), with specifications detailed in Appendix C.1. Figure 9d shows scaling plots for SiT-$\{S, B, L\}$ and U-Net-$\{XS, S, B, L\}$, with each model evaluated at $\{200K, 300K, 400K\}$ training iterations.

## C.9    Verification of Perception-Aligned Training Hypothesis (Section 5.2)

In this section, we verify our central Hypothesis **C4** proposed in Section 5.2. Using a carefully designed 2D toy dataset, we illustrate how PAT–*aligning a neural network with perception*–leads

to *better extrapolation*, and, consequently, perceptually better samples. Importantly, models are trained on a tiny training set $\mathcal{D}$, making *supervision nearly perfect* (i.e., models fit the training region almost exactly), which allows us to study *extrapolation in isolation*. In this setting, in Equation (5), the supervision loss is fixed at $\mathcal{L} \approx 0$, so generative performance (perceptual quality of samples) directly reflects the extrapolation function $f_{\text{extrapolation}}$.

**Setup.** Our 2D toy data is designed to be *analogous* to image data, as shown in Figure 11.

(S1) Training data $\mathcal{D} := \{▲(-1,0), ▲(-0.2,0), ■(0.2,0), □(1,0)\}$, small enough to fit perfectly.

(S2) Two points are Euclidean close if their $(x,y)$-distance is small (e.g. ▲, ■). We (arbitrarily) define two points as *perceptually close if angular distance is small* (e.g. ▲, ▲ or ■, □).

(S3) As shown in Figure 8 (middle), "bad" samples interpolate between Euclidean-close data (▲ – ■), while "good" samples interpolate between perceptually close data (▲ – ▲ or ■ – □).

**Implementation of PAT.** Based on this setup, we describe the implementation of PAT Instances. For the baseline (without PAT), we use a sufficiently large 2-layer MLP, except for the last instance.

(P1) *Aligning diffusion space*: Instead of the default Cartesian space $(x,y)$ (non-perceptual), we train a model in polar space $(r, \cos\theta, \sin\theta)$, emphasizing the angular (perceptual) component.

(P2) *Aligning representation*: Since directly manipulating a neural net's internal representation is nontrivial, we instead train kernel ridge regression models with a Gaussian kernel. The baseline uses non-perceptual features $(x,y)$, while PAT uses perceptual features $(r, \sin\theta, \cos\theta)$.

(P3) *Aligning architectural bias*: Analogous to translational equivariance in image convolutions, we use a radial-equivariant MLP as a stronger inductive bias.

**Results.** Generated samples are shown in Figures 9b to 9d: baseline (left) vs. PAT (right).

(R1) Baselines generate "bad" samples by interpolating between Euclidean-close points, due to the lack of perceptual bias. In contrast, all PAT instances generate "good" samples by interpolating between perceptually close points. As supervision is kept near-perfect, this verifies **C4**: *PAT, purely by improving extrapolation, enables generating dramatically better samples*.

(R2) All three PAT implementations yield similarly shaped sample distributions, showing that, despite methodological differences, they operate on the same underlying principle.

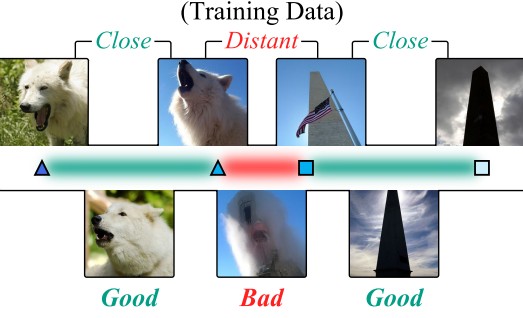

Figure 11: **Comparison of "good" vs. "bad" samples** on ImageNet, illustrated with an analogy to a 2D toy example. Toy training data: (▲, ▲, ■, □); generated samples: ▬, ▬.

