# OpenReview forum: "Selective Underfitting in Diffusion Models"
_ICLR.cc/2026/Conference — Submitted to ICLR 2026_

### Official Review · Reviewer_X3ZP · 2025-10-30

**Soundness:** 4
**Presentation:** 4
**Contribution:** 2
**Rating:** 6
**Confidence:** 3

**Summary:**

The paper investigates how well a trained diffusion model approximates the empirical score. In particular, the paper challenges the common perspective (Perspective A) that a perfectly trained diffusion model should approximate well the empirical score \textit{everywhere}, i.e., over the entire data space. A key observation made by the authors is that the supervision signal, at a certain noising step $t$, coming from the DSM objective, only covers a relatively small shell around each noised training data point, leaving a large part of the space unsupervised. This results in the model overfitting the empirical score well in these limited regions (Supervised regions) and underfitting it outside them (Extrapolated regions). As a consequence, at generation time, the model will most likely explore regions for which very little supervision signal has been received during training, hence requiring it to extrapolate. Interestingly, the experiments show that too much supervision signal leads the model to memorise more the training data. On the other hand, giving the model more freedom to extrapolate significantly reduces this effect. Furthermore, if the training objective is modified to shape the model's behaviour in the extrapolation regions (through inductive biases such as convolutions) can lead the model to more favourable performance during generation.

**Strengths:**

- The paper is clearly written and well structured. The exposition follows a natural flow and claims are substantiated with adequate evidence.
- The observation that the empirical score cannot be precisely approximated everywhere represents -- from the best of the reviewer's knowledge -- a simple yet relatively novel perspective and has important implications on the generation quality and on memorisation.
- I find experiment 4.3 interesting: counterintuitively adding more supervision data actually increases the effect of memorisation, pointing to the fact that the extrapolation behaviour of the model plays a major role in generalisation.

**Weaknesses:**

- The main weakness of the paper is that it does not discuss in depth the behaviour of diffusion models in the extrapolation region and which factors mostly influence it. This seems a natural (and crucial) question given that in high-dimensions the supervised region is very small and the behaviour of the model outside it is undefined a priori.
- While the perspective taken by the paper is somewhat novel, there is a large overlap with prior work (see for instance [1,2,3] and reference therein), mainly on memorisation in diffusion models, that is not discussed.

[1] Sclocchi, Antonio, Alessandro Favero, and Matthieu Wyart. "A phase transition in diffusion models reveals the hierarchical nature of data." Proceedings of the National Academy of Sciences 122.1 (2025): e2408799121.

[2] Biroli, G., Bonnaire, T., De Bortoli, V., & Mézard, M. (2024). Dynamical regimes of diffusion models. Nature Communications, 15(1), 9957.

[3] Achilli, Beatrice, et al. "Memorization and generalization in generative diffusion under the manifold hypothesis." Journal of Statistical Mechanics: Theory and Experiment 2025.7 (2025): 073401.

**Questions:**

- On the links with prior work: Previous works have studied the emergence of memorisation in backward diffusion (see for instance [1,2,3] among others). Some of these works have also investigated the onset of memorisation as a function of denoising time.  How does the present work link to these findings?
- On the role of inductive biases: In the introduction the authors state "We advance a mechanism for how diffusion models generate
samples beyond their training data that is agnostic to architectural inductive biases, unlike previous theories". I would appreciate if the authors could clarify what they mean in this statement. In particular, from my understanding of the paper, inductive biases are shown to ultimately play a major in shaping the behaviour of the model in the extrapolation region, where little supervision is provided.

---

### Official Review · Reviewer_4Qh5 · 2025-10-30

**Soundness:** 1
**Presentation:** 3
**Contribution:** 2
**Rating:** 2
**Confidence:** 3

**Summary:**

This paper studies underfitting and generalization in diffusion models. The central claim is that during training, noisy samples $x_t = \alpha_t x_0 + \sigma_t \varepsilon$ concentrate on thin shells around training examples (“supervised regions”), leading the model to approximate the empirical score $s_\star(\cdot, t)$ there. Outside these regions (“extrapolation regions”), the learned score $s_\theta(\cdot, t)$ deviates more from $s_\star$, i.e., **selective underfitting**.

Empirically, the authors (i) investigate supervised regions on ImageNet and show they remain largely non-overlapping until relatively large noise levels ($t \approx 0.8$); (ii) quantify selective underfitting via score-error gaps; and (iii) use noise inversion and a distillation experiment to argue that trajectories entering supervised regions tend to yield training examples. They further propose practical implications (e.g., predicting final FID; interpreting RePA/architectural effects) via their underfitting metric.

**Strengths:**

1. The observation that supervised regions are largely disjoint up to $t\approx 0.8$ is interesting and consistent with prior reports that learned vs. empirical scores diverge at similar timesteps.
2. Figures (e.g., Figs. 1 and 5) are intuitive, and the paper is well formatted.

**Weaknesses:**

1. “Selective underfitting” is essentially evaluating the **training/empirical denoising loss** on test data. It is well known that a generalizing denoiser deviates from an empirical denoiser in such regions and can denoise a test image rather than overfit. Rather than being a key to generalization, selective underfitting reads more like a *necessary* or descriptive condition (“where the model is not fitting the empirical score”) than a causal/mechanistic explanation of *how* extrapolation succeeds.
Moreover, there are more directional, theory-grounded probes of generalization than the proposed gap metric—for example, test denoising loss in [1].
2. The selective underfitting framing also struggles to reconcile several relevant findings:
   1. Reproducibility that co-arises with generalization in diffusion models [1].
   2. The memorization→generalization transition as the number of training samples increases [2].
   3. Evidence that the learned score deviates from the empirical score even on supervised regions, and that such deviation can *enable* generalization (see Fig. 2 in [3]).
3. **Issues with experiments.**
   1. Strongly overfitting (memorizing) models can still learn imperfect empirical scores, but still generate training data. Additional study and comparison on the selective underfitting of such memorized models would strengthen the claims.
   2. Mismatch in Fig. 2: Comparing score error on training-time forward noisy samples (blue) vs. inference-time backward states (red) is not fully fair. A better test is to complete sampling to obtain a novel image, then add noise at the same $t$ and re-measure score error on that generated sample.
   3. Results in Fig. 3: The difference between supervised (≈$1.8\times10^{-1}$) and extrapolated (≈$5.4\times10^{-1}$) errors is reported, but without a standardized baseline it is hard to judge (i) how much the model actually *overfits* in supervised regions and (ii) whether the extrapolation error is “large” in a practically meaningful sense.
   4. Results in Fig. 4:The noise-inversion experiment does not directly establish memorization or generalization; it shows a property of trajectories relative to supervised regions.
   5. Section 5: Several conjectures are offered with minimal evidence (two teasers with toy examples) and feel disconnected from the core underfitting analysis.

**Questions:**

1. **Definition of supervised regions.** Are they strictly the thin spherical shell (as in Prop. 3.2), or the full ball around each sample? In my experiments, the network behaves as a strong “blind” denoiser and can denoise with *mismatched* noise levels [4], which would enlarge the effective supervised region.
2. **Distillation setup (Fig. 5).** Fixing $D_{\text{score}}$ and enlarging $D_{\text{region}}$ may force the model to project many samples toward a few anchor images in $D_{\text{score}}$, just like effective training-data duplication, which naturally promotes memorization. Why not directly minimize MSE on a *thickened* shell (by perturbing the noise level) rather than distilling $s_\star(\cdot)$ over a larger range of data (i.e., addinging more supervised shells)?
3. These works on interpolation and local overfitting may be useful to connect with [5, 6].

*My current rating is a provisional assessment and can be updated after author responses and discussion.*

**References**

 [1] Kadkhodaie et al., *Generalization in Diffusion Models Arises from Geometry-Adaptive Harmonic Representations*, ICLR 2024.
 [2] Zhang et al., *The Emergence of Reproducibility and Consistency in Diffusion Models*, ICML 2024.
 [3] Niedoba et al., *Towards a Mechanistic Explanation of Diffusion Model Generalization*, ICML 2024.
 [4] Sun et al., *Is Noise Conditioning Necessary for Denoising Generative Models?*, ICML 2025.
 [5] Zeno et al., *How Do Minimum-Norm Shallow Denoisers Look in Function Space?*, NeurIPS 2023.
 [6] Zeno et al., *When Diffusion Models Memorize: Inductive Biases in Probability Flow of Minimum-Norm Shallow Neural Nets*, arXiv.

---

### Official Review · Reviewer_eMqt · 2025-11-02

**Soundness:** 2
**Presentation:** 2
**Contribution:** 2
**Rating:** 2
**Confidence:** 4

**Summary:**

This paper studies where diffusion models underfit the empirical score, introducing the concept of **selective underfitting** to describe spatially non-uniform score approximation. The authors observe that while training samples lie in narrow Gaussian shells around data points (the denoising targets), inference trajectories often move beyond these shells into regions rarely supervised during training. This leads to a mismatch between training and inference distributions. The paper (1) empirically demonstrates selective underfitting in two theoretically grounded regions of data space, (2) proposes a mechanism explaining how diffusion models can still generate novel samples beyond training data independent of architectural inductive biases, and (3) introduces a scaling law quantifying the efficiency of training procedures, offering a unified perspective on diffusion model optimization and generalization.

**Strengths:**

**1. Novel conceptual framework:** The idea of selective underfitting provides a compelling and fine-grained understanding of diffusion model training and generalization, refining prior uniform-underfitting views.

**2. Well-written and clearly structured:** The paper is clearly written, with strong motivation, intuitive figures, and well-organized arguments that make a complex topic accessible. The exposition effectively connects conceptual insights with empirical evidence, enhancing readability and comprehension.

**Weaknesses:**

Here are some main concerns about this paper.

**1. The paper does not fully engage with the essence of underfitting in diffusion models.** While the notion of selective underfitting provides a spatially structured view of how diffusion models approximate the empirical score, it does not fundamentally revisit why underfitting occurs in the first place. Recent works have provided deeper theoretical explanations of this phenomenon. Specifically, *Kadkhodaie et al. (2023)* argue that generalization in diffusion models arises from geometry-adaptive harmonic representations, where the learned score function captures the intrinsic geometry of the data manifold rather than merely fitting local noise perturbations. Similarly, *Bonnaire et al. (2025)* show that diffusion models avoid memorization due to implicit dynamical regularization in the training dynamics, which naturally biases the model toward smoother score fields aligned with the true data distribution.

In light of these perspectives, the present paper’s “selective underfitting” can be viewed more as a manifestation or empirical measure of generalization rather than a fundamentally new explanatory mechanism. The authors discuss where underfitting occurs but not why diffusion models underfit in a principled manner. As a result, the proposed framework risks being descriptive rather than mechanistic—it highlights the spatial distribution of score errors but leaves the underlying regularization or representation principles unexplored.

1. Kadkhodaie, Zahra, et al. "Generalization in diffusion models arises from geometry-adaptive harmonic representations." arXiv preprint arXiv:2310.02557 (2023).
2. Bonnaire, Tony, et al. "Why Diffusion Models Don't Memorize: The Role of Implicit Dynamical Regularization in Training." arXiv preprint arXiv:2505.17638 (2025).

**2. Heavy reliance on conjecture and empirical observation, with limited principled understanding.** The paper relies heavily on conjecture and empirical observation, offering limited theoretical explanation for why selective underfitting emerges. While the experiments are suggestive, the arguments remain largely descriptive rather than derived from score-matching principles or diffusion dynamics. As a result, the work provides intuition but not a principled understanding of the underlying mechanisms.

**Questions:**

The paper links selective underfitting to the generative (sampling) behavior of diffusion models, but it is unclear why underfitting during score learning should directly affect the reverse sampling process. Since overfitting or underfitting is determined primarily by the training dynamics of the score network, not by the stochastic sampling procedure, could the observed sampling behavior simply reflect how the model extrapolates from its learned score field rather than underfitting per se? How do the authors conceptually or empirically separate the effects of training-time underfitting from those of the reverse diffusion dynamics?

**Details Of Ethics Concerns:**

N.A.

---

### Meta-Review · Area_Chair_jQns · 2026-01-06

**Summary:**

The submission presents an empirical study on the hypothetical factors that affect generalization behavior of diffusion models. The reviewers acknowledge the clear presentation and some insights behind the hypothesis and the experimental design. They also expressed a few concerns regarding the core contribution:
1. The study lacks setting up a mechanism where underfitting would occur before carrying out the experiments, limiting its value in transitioning into a deeper understanding of the working principle of diffusion model generalization.
2. Rationale of the assumption still appears unclear. Particularly, Reviewer 4Qh5 mentioned that a perfectly overfitting model, which generates training data points, still underfits in the extrapolation regions since no supervision exists there.
3. On the generalization of diffusion models, there are quite a few works that have reported solid findings and conclusions. This submission does not provide a sufficient discussion on the connection and advancements over these existing works.
4. There exist issues in the experiment design and result presentation, making the claims not sufficiently solid.

**Reviewer Concerns:**

The authors did not provide any rebuttal. Up to my judgement, I would perceive the mentioned concerns as reasonable challenges on the quality and implicated contribution of this work. Particularly, regarding the first two concerns, in addition to reviewers' challenges, I would also wonder a more formal definition of supervision region and extrapolation region (also mentioned by Reviewer X3ZP) given that the input to the model in training is probabilistic and can take any position in principle; even only considering highly-probable regions, although the model may be underfitted outside, the model would hardly encounter those samples in inference. Although I typically would not put a major weight on missing discussion with related work, in the case of this submission, this turns out to be an insufficiency in assuming a clear and proper mechanism.

**Reviewer Scores:**

The authors did not provide any rebuttal.

---

### Decision · Program_Chairs · 2026-01-26

Reject